# The Neonatal Microbiome: Implications for Amyotrophic Lateral Sclerosis and Other Neurodegenerations

**DOI:** 10.3390/brainsci15020195

**Published:** 2025-02-14

**Authors:** Andrew Eisen, Matthew C. Kiernan

**Affiliations:** 1Division of Neurology, Department of Medicine, University of British Columbia, Vancouver, BC V6T 1Z4, Canada; 2Neuroscience Research Australia, University of New South Wales, Randwick, Sydney, NSW 2031, Australia; matthew.kiernan@neura.edu.au

**Keywords:** amyotrophic lateral sclerosis, microbiome, neonatal, excitation/inhibition

## Abstract

Most brain development occurs in the “first 1000 days”, a critical period from conception to a child’s second birthday. Critical brain processes that occur during this time include synaptogenesis, myelination, neural pruning, and the formation of functioning neuronal circuits. Perturbations during the first 1000 days likely contribute to later-life neurodegenerative disease, including sporadic amyotrophic lateral sclerosis (ALS). Neurodevelopment is determined by many events, including the maturation and colonization of the infant microbiome and its metabolites, specifically neurotransmitters, immune modulators, vitamins, and short-chain fatty acids. Successful microbiome maturation and gut–brain axis function depend on maternal factors (stress and exposure to toxins during pregnancy), mode of delivery, quality of the postnatal environment, diet after weaning from breast milk, and nutritional deficiencies. While the neonatal microbiome is highly plastic, it remains prone to dysbiosis which, once established, may persist into adulthood, thereby inducing the development of chronic inflammation and abnormal excitatory/inhibitory balance, resulting in neural excitation. Both are recognized as key pathophysiological processes in the development of ALS.

## 1. Introduction

Amyotrophic lateral sclerosis (ALS/MND) is a complex, uniquely human neurodegenerative disease with a variety of phenotypes, including frontotemporal dementia [1,2,3]. ALS phenotypes (age of onset, site of initial clinical presentation, and disease duration, amongst others) are predicated by genetic, environmental, lifestyle, and epigenetic influences [4]. There are no known naturally occurring animal models, and induced animal models whilst usefully mimicking anterior horn cell death, and to a lesser extent loss of upper motor neurons [5], cannot truly recapitulate all aspects of the disorder as seen in humans [6,7]. Many consider ALS to be a primary brain disorder [8].

Neurodegenerations, including amyotrophic lateral sclerosis (ALS), have increased over the past two centuries [9,10]. This time period is short, encompassing a limited number of generations, making genetic factors alone an unlikely explanation [11]. Post-industrial revolution changes in lifestyle and environmental factors compared with conditions experienced during the preceding evolutionary period are relevant to the increasing incidence of ALS and other neurodegenerations [12,13]. A steady decline in dietary fruits, vegetables, and fibers and increased consumption of animal products, saturated fats, and refined sugars has exerted evolutionary pressure on the gut microbiota [14,15] and adversely influenced the metabolic and inflammatory profile of brain cells [16]. Dietary and broader lifestyle and environmental changes together with increased longevity have further impacted the human microbiome [15,17,18].

The microbiome consists of bacterial, archaeal, fungal, viral, and microscopic eukaryotes and protozoan communities that colonize multiple body sites. They form an interface between the host and the outside world through the gastrointestinal tract, skin, respiratory tract, and urogenital tract [19,20]. Most of the microbiome is contained within the gastrointestinal tract, and the total genome of the organisms colonizing it is many times greater than that of the human genome [21,22].

Imbalance of the gut microbiome and dysfunction of the gut–brain axis may develop because of diet, metabolism, altered immunity, age, stress, lifestyle, antibiotics, and other therapeutic agents [11,18]. The effect of dysbiosis appears greatest in early life, particularly during “the first 1000 days”, which spans conception to a child’s second birthday [23]. During this period of maturation, the gut and brain interact, with the gut microbiome playing a key role in shaping processes that support neural health [24,25,26].

Inappropriate signals within the gut–brain axis may induce low-grade inflammation, oxidative stress, disturbed energy metabolism, impairment of the blood–brain barrier, and increased cellular aging [11,16,27,28,29]. These are shared pathophysiological mechanisms involved in all neurodegenerations including ALS [30,31]. Neurodegenerations may exhibit manifestations long before classic features appear [32,33]; this too is true of sporadic ALS [34]. This includes gastrointestinal symptomatology related to dysbiosis occurring prior to the onset of typical Alzheimer’s disease, Parkinson’s disease, and ALS [35,36,37]. But it is possible that sporadic ALS has its origins in early life [38]. The “First 1000 Days of Life” are critical to brain development, encompassing synaptogenesis, myelination, neuroplasticity, and the formation of functional neuronal circuits. Successful completion of these fetal and neonatal brain development processes may be impacted by a variety of processes, including the initial colonization and establishment of the gut–brain axis [39]. As such, the microbiome represents an essential environment that links the processes of human physiology, metabolism, and the immune system [40] during this vital period of neuro-development [41,42,43,44,45], with alterations potentially promoting future neurodegenerations such as ALS [46,47,48,49,50,51]. In further support of such concepts, there is growing recognition that early-life dysbiosis may be causative of later-life neuro-psychiatric diseases [52,53]. With these background concepts, the present review aims to explore current evidence that may suggest a role for the microbiome as a potential contributor to the origin of ALS and other neurodegenerations, most particularly during the perinatal period [38]. ALS is viewed as a multistep process in which the first step includes the genome, in utero, and maternal influences [54,55,56,57]. The maturation of the microbiome might be considered an early life contributor to this multistep process.

## 2. Maturation of the Microbiome

While adults have relatively stable gut microbiomes, the developing neonatal microbiome is unstable, with greater adaptability. At birth and through the initial years of life, the microbiome composition changes and expands [58]. Its composition undergoes rapid evolution during the first 1000 days of life, [59] with changes related to altered modes of delivery [60,61], early life nutrition [62], and deficiencies in maternal diet such as folate, iron, or omega-3 fatty acids with a potentially negative impact on mitochondrial function [63] and antibiotic exposure [64] (Figure 1).

Similarly, microorganisms begin to colonize the skin [65] and mucosal cavities (oral [66], nasal [67], vaginal [68,69], and pulmonary [70,71]). However, the greatest colonization occurs within the gastrointestinal tract [72]. Microbial exposure precedes conception and gestation within the male and female reproductive tracts [73] and maturation in the fetal gut microbiome starts in utero [22,74].

At this stage, data from studies of fetal microbiome composition remain inconsistent [75], with controversy as to the existence of a placental microbiome [76]. Nevertheless, the maternal microbiome is implicated in placental structure and function [77,78]. The maternal microbiome drives the maturation of the offspring’s gut microbiome by means of local transmission during birth, with further exposure during maternal diet and breastfeeding [69,79]. It is accepted that breast milk contains important bioactive components that affect the establishment of the infant microbiome, including immunoglobulins, oligosaccharides, complement lactoferrin, lysozymes, hormones, and cytokines [80,81]. Human milk oligosaccharides are essential bioactive constituents and serve as prebiotics, mucosal signalling agents, and immunomodulators and significantly contribute to the enrichment of the gut microbiota, the enhancement of intestinal epithelial barrier integrity, and the support of immune function [81]. Brain development and the gut microbiota co-evolve, suggesting a bi-directional influence between the brain and commensal bacteria [82], while the developing gut–brain axis impacts the physiological and structural development of the central nervous system [83,84]. In terms of this bidirectional influence, the change from breast feeding to a solid diet is pivotal to the maturation of the gut flora [85,86], occurring at a time that coincides with intense synaptogenesis within the brain [79,87,88,89,90].

## 3. Brain Development and Gut Microbiome

The adult gut microbiome has diverse functions (see the review by Nandwana et al. [91]). During the “First 1000 days”, bacteria may exert an influence on the fine-tuning of synaptogenesis and the formation of neuronal circuits. In turn, neuromodulation is altered by chemical exchange through the gut–brain axis, manipulated via the vagal nerve, immune signalling, and bacterial production of metabolites [92]. Microbiota signalling also contributes to myelinization, especially in the prefrontal cortex [52,53,82,93].

Of further relevance, a large variety of metabolites are produced within the gut lumen, including neurotransmitters (gamma-aminobutyric acid, serotonin, dopamine, acetylcholine, and noradrenaline), several vitamins, short-chain fatty acids (propionate, acetate, and butyrate), and amino acids and their derivatives [94]. After absorption, some of these molecules cross the blood–brain barrier to reach and influence brain function [95]. But the mechanisms underlying the complex interactions of gut-produced metabolites remain to be determined [43,96]. Similarly, dysbiosis increases intestinal permeability with the translocation of harmful microbial products into the bloodstream that are able to promote an inflammatory response and damage the blood–brain barrier.

## 4. Neonatal Gut Microbiome and Immunity

The adult central nervous immune system is specialized and tightly regulated. Primary immune cells include microglia, astrocytes, T cells, and B cells, responding together to phagocytose debris, release cytokines, and recruit other immune cells. The immune responses are balanced to prevent excessive inflammation that induces neuronal damage and neurodegenerative diseases.

Maternal antibodies can be transferred placentally prior to birth. This passively derived immunity protects for the first few months of life through the transfer of IgG antibodies [97]. After birth, maternal milk provides the first source of antibody-mediated protection in the intestinal tract of infants against infection. In newborns, the immune system undergoes a rapid transition from dependency on maternal protection to becoming self-sufficient. Maternal microbes are transferred during delivery, then through maternal milk [98], and later when weaning to solid food occurs. Each of these steps advances colonization with microbe adherence to the intestinal epithelium [99] and maturation of the immune system [97]. The immune system becomes educated and expanded through infancy and early childhood through interactions with the gut microbiota [100], and if dysbiosis occurs during early development, it substantially impairs immune system elaboration [101].

Different signals and mechanisms derive from the developing microbiome, which controls immune activation [102]. This includes microbiome epigenetic remodelling and altered gene expression, the production of tissue-protective anti-inflammatory cytokines (e.g., IL-10), and curtailment of pro-inflammatory cytokines (e.g., IL-6 and TNF-α), which prevents excessive inflammation. Inflammatory control is mediated through microglia, the immune cells of the brain parenchyma [103]. But glias’ role extends much beyond immune functions [104]. They also sense neuronal activity, regulate neuronal synaptic pruning [105,106], affect developmental patterning and homeostatic functions in the central nervous system, including cell and/or debris clearance, synaptic maturation, neural circuit function, angio-/vasculogenesis, myelination, neurotransmission, and help maintain the integrity of the blood–brain barrier [103].

## 5. Neonatal Dysbiosis Induces Neurodegeneration

There are different but interacting mechanisms through which dysbiosis arising in the neonatal microbiome may affect neurodevelopment, with the subsequent potential for later-life neurodegenerative disease [31,107] (Table 1). Given that the neonatal gut microbiome is critical to training the immune system during early development [79,108,109], dysbiosis during this period can impair immune regulation, thereby causing overactivation of inflammatory pathways and resulting in low-grade chronic inflammation. Chronic inflammation activated through pro-inflammatory cytokines such as IL-6 and TNF-α is a major contributor to processes linked to ALS and related neurodegenerations [110,111,112,113,114], which may in turn suggest novel considerations for therapy [115].

At a pathological level, mislocalization and aggregation of the TAR DNA binding protein 43 (TDP-43) within the cytoplasm of neurons and glia is a pathological hallmark of ALS [116,117,118]. Growing evidence has associated neuroinflammation immune-mediated mechanisms with TDP-43 toxicity [119]. Cytoplasmic aggregates of TD for P-43 have also been implicated in neuronal excitotoxicity [120], considered a prime pathogenic mechanism in ALS [121,122,123]. Interleukin-1beta (IL-1beta) is a proinflammatory cytokine that contributes to the pathogenesis of both acute and chronic neurological disorders and mechanistically links the pro-inflammatory response to glutamate excitotoxicity [124]. TDP-43 increases blood barrier permeability and leukocyte recruitment, indicating complex intermolecular interactions between systemic inflammation and pathological TDP-43 protein, promoting disease progression [125].

There is scientific support for the role of the gut microbiome in preserving the integrity of the gut barrier [126,127]. Gastrointestinal barrier dysfunction (“leaky gut”) contributes to the development and progression of chronic low-grade systemic inflammation and age-related diseases such as ALS [128]. Gut bacteria may release metabolites into the blood that readily cross the blood–brain barrier [95]. Most gut microbes reside within the intestinal lumen lined by epithelial cells. Disruption of this gut epithelial barrier, as caused by pathogens, allows unregulated translocation of microbes into the lamina propria where gut immune cells reside [129].

## 6. Neurotransmitters and the Excitatory/Inhibitory Balance

Resident gut microbes, particularly bacteria, produce and utilize a variety of chemical messengers, including neurotransmitters critical for communication with the nervous system [130]. They include dopamine (metabolized through tyrosine), noradrenaline, serotonin (converted through tryptophan), GABA, and acetylcholine [131]. The pro-excitatory neurotransmitter glutamate and anti-excitatory neurotransmitter gamma-aminobutyric acid (GABA) plays a crucial role in regulating neuronal excitability [132]. Glutamate can be acquired from the diet and eukaryotic cells, synthesized by the microbiome, and can also be converted into GABA [132]. As demonstrated in animal studies and in humans, the manipulation of bacterial neurotransmitters impacts host physiology [94]. Dysbiosis may adversely alter or reduce neurotransmitter production, with a variety of deleterious secondary effects [133].

Of the variety of neurotransmitters produced by bacteria, GABA may be the most crucial because of the role of GABAergic inhibitory circuitry in excitatory/inhibitory (E/I) balance [134]. GABA is one of the earliest and most highly evolutionary conserved neurotransmitters [135,136]. During development, GABA is excitatory, while embryonic GABA signalling is the main excitatory drive for developing cortical networks [137,138,139]. The ability of embryonic GABA to depolarize primitive neurons arises because of a high intracellular chloride concentration. Excitatory GABAergic neurons migrate into the cerebral cortex via the white matter throughout the second half of gestation, and the switch to the typical adult inhibitory phenotype is greatest during the first postnatal year [140]. However, the change from excitatory (E) to inhibitory (I) continues to mature for several years before it is complete. Impaired maturation of E/I balance, particularly during embryogenesis, can exert lasting effects [134,141]. Maturation of E/I balance normally results in a decrease in overall excitatory tone in concert with sophisticated inhibitory control of neural activity. Failure of this normal maturation results in a net excitation recognized as a key pathophysiological factor driving the development of ALS [136].

The predominant excitatory neurotransmitter glutamate is essential for maintaining the metabolic performance of neurons and glia and the maintenance of proper E/I balance. After taking up glutamate via excitatory amino acid transporters (EAAT1 and EAAT2), astrocytic glutaminase hydrolyzes glutamate to glutamine, which is then transported to neurons [142]. Overall evidence points to loss of inhibition rather than excess glutamatergic excitation as the main driver behind ALS excitotoxicity [123,143,144].

## 7. Changes in Brain Permeability

Gut microbiome metabolites are important in the formation of the BBB during embryonic and neonatal life [145]. For example, the microbiome-derived fat-soluble vitamin K2 can diffuse freely across the BBB and regulate a wide spectrum of molecular mitochondrial functions [146]. During pregnancy, the maternal gut microbiota regulates the developing fetal BBB by upregulating the expression of proteins such as claudin-5 [29]. Claudin-5 (CLDN5) is a protein that helps form tight junctions, particularly in the blood–brain barrier (BBB) [147]. It is a key component of the BBB cell membrane and is involved in regulating the permeability of the barrier. Microbiome-induced inflammation can alter the permeability of the BBB, disrupting the tightly packed endothelial lining of the capillaries supplying the brain [148]. Proteins sealing the gaps between brain vascular endothelial cells are critical to its regulation [149,150], and these same molecules mediate intestinal permeability [151].

Several bacterial metabolites can cross the BBB through various transport mechanisms and accumulate in the brain, directly impacting its function. Detrimental gut–brain interactions may occur when intermediaries including bacteria, toxic digestive metabolites, bacterial toxins, and other virulent factors such as cytokines ‘leak’ into the bloodstream [152]. For this to happen, they must cross through both the intestinal epithelium barrier and the blood–brain barrier (BBB). Altered permeability of the intestinal epithelium results from a dysbiotic gut microbiome, which damages intestinal epithelial cells and facilitates the translocation of gut microbiota across the lumen to the mesenteric lymph and peripheral circulation. Bioactive metabolites that breach the BBB may impair the regulation of mitochondrial oxidation and microglia activation and induce pathogenic protein aggregation, important steps in the pathophysiology of neurodegeneration [132].

## 8. Mitochondrial Dysfunction and the Microbiome

Mitochondria are the metabolic hubs underlying a wide range of cellular processes, with mitochondrial dysfunction linked to the processes of neuronal death in ALS [153]. Mitochondria and the microbiome are closely related through their shared evolutionary background, maternal inheritance patterns, and overlapping roles in the maintenance of health and disease (“mitochondria-microbiome crosstalk”) [154]. Of relevance, increased levels of reactive oxygen species (ROS) may potentially damage DNA, lipids, and proteins, thereby contributing to age-related neurodegenerations, with the interplay between mitochondria and gut microbes through reactive oxygen species (ROS) [155]. Mitochondrial dysfunction caused by dysbiosis represents a major contributing factor to disruption of the gut epithelial barrier, with the potential for subsequent impairment of blood–brain barrier permeability, leading to neuroinflammation [156,157].

## 9. Future Considerations

Defective maturation of the neonatal gut microbiome and gut–brain axis may represent an initial brain insult, with only subtle deviations from a normal neonatal microbiome required to initiate a trajectory of chronic inflammation and excitotoxicity, both major contributors to the ultimate molecular cascade resulting in clinical ALS [123,158]. Both immunity and inflammation relate specifically to TDP-43, whose mislocalization and aggregation are hallmarks of ALS [119]. Over a lifetime, other interacting factors contribute to the effects of inflammation, and Betz cell neurons and the corticomotoneuronal system have been proposed as the nidus of origin of ALS and they are particularly vulnerable to inflammatory and excitotoxic insults and effects of aging [8]. The adult microbiome is readily manipulated through diets designed to introduce specific beneficial strains of bacteria or fecal microbiota transplants. However, the potential therapeutic benefits of these approaches in neurodegenerations, including ALS, have fallen short (see the review by Loh et al. [159]). Such procedures are not without risk, especially during early neurodevelopment [160,161]. Since most studies have been undertaken in rodent models, similar effectiveness does not necessarily translate to human disease [162].

There are further evolutionary considerations, given that the gut microbiota of hunter–gatherers and populations consuming a rural agrarian diet harbor greater diversity than the microbiota of the modern Western world [163,164]. Humans have experienced major dietary changes from gathered to farmed foods to the now mass consumption of processed meals. Each dietary shift has been accompanied by a concomitant adjustment in the microbiota and has induced a loss of microbiota diversity, postulated to be magnified over generations [163]. Much of the vast biochemical potential of the microbiome is distinct from that of the host, with the ability to modify phenotypes where host organisms gain new physiological abilities as a result of contributions from their microbial partners [165].

The idea that commensal microbiota can modulate the expression of the human genome is new, with information derived through the introduction of microbes into germ-free animals [166]. Intestinal gene expression undergoes dramatic reprogramming with microbial colonization after birth [167]. The microbiome encodes many more genes than the host genome, and interactions with it may alter the host genotype–phenotype [168]. Combining the microbiome’s consortium of genomes extends the genetic repertoire of the host, forming an “Extended Genotype” [168,169]. Wilde et al. [170] reviewed the host control of the microbiome in-depth, stressing that the generation time of symbionts is typically extremely short relative to their host’s, which enables rapid shifts in the species composition of microbiotas. Over evolutionary history, the incorporation of new species into the microbiota may have allowed a host species to take behavioral and social leaps, often ascribed to factors like altered brain morphology [165], and adaptations are most evident when microbes are first introduced in the neonatal period [171]. Technical improvements will allow for in-depth analysis of the microbiome and host genomes and how they influence each other [172].

It has been proposed that ALS reflects a multistep process in which a series of biological processes result in a tipping point where the disease becomes manifest [54,55,56,57,173]. There may be six distinct processes, but the number is variable; for example, if there is a large effect mutation, there would be fewer steps [55]. The steps include not only genetic and epigenetic factors but also environmental exposures, as encompassed in the ALS exosome [12]. An unrecognized step of the multistep process may relate to the neonatal microbiome and its interactions with the host, considered another clue to the causation of neurodegenerations, and most specifically ALS [2]. Important to such a consideration, there remain potential therapeutic strategies to modulate the gut microbiota, including antibiotics and probiotics, in addition to nutritional interventions that may support the function of beneficial microbes, and microRNAs to enhance targeted microbial populations [174]. Clearly, such considerations should be formally evaluated through clinical trials, through a precision medicine approach, with biomarker evaluation [13].

## 10. Conclusions

Early life environmental influences may have a profound impact on neurodevelopment and subsequently be a factor in the development of neurodegeneration, including ALS [113,175]. One such environmental factor is the gut microbiota, which, immediately after birth, rapidly and densely populate the newborn with complex forms of microbes. As we have described above, the effects of the maturing microbiome are diverse and have an influence on many neurobiological processes that may be impaired by dysbiosis during its maturation. Modern lifestyles, including refined diets, antibiotic intake, exposure to air pollutants, microplastics, and stress all negatively affect the diversity and composition of the gut microbiota [15]. Any combination of these factors during the first 1000 days, including pregnancy, delivery, and neonatal life, may result in infant dysbiosis, which, if unattended, becomes a potential risk factor for ALS and other neurodegenerations. Further confirmation of this concept is important as implicating very early-life dysfunction as a clue to later-life neurodegenerations opens a large time scale of possible interventions.

## Figures and Tables

**Figure 1 brainsci-15-00195-f001:**
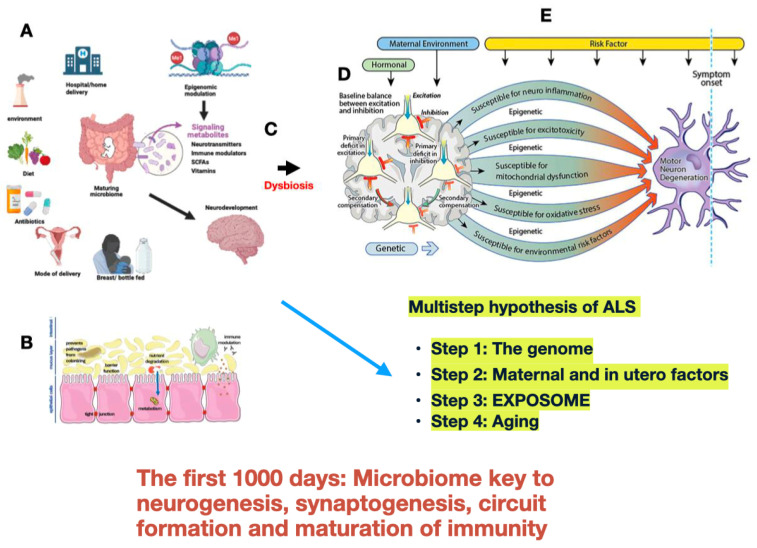
Modified from Kiernan M.C., Ziemann U., and Eisen A. Amyotrophic lateral sclerosis: Origins traced to impaired balance between neural excitation and inhibition in the neonatal period [38]. Various factors influence the maturation of the fetal/neonatal microbiome (**A**). Establishment progresses rapidly during delivery, breastfeeding, and with the institution of a diet. Metabolites (neurotransmitters, vitamins, short-chain fatty acids, and amino acids) produced in the gut are key to neurodevelopment through a variety of neurobiological processes chiefly acting via the gut–brain axis and are modulated through epigenetic interaction which determines individual sensitivity and susceptibility. (**B**). Neonatal dysbiosis can set the stage for progressive neuroinflammation, excitotoxicity, mitochondrial dysfunction, and excessive oxidative stress to which motor neurons are susceptible (**C**). During neurodevelopment, neural network establishment and functioning are sensitive to excitatory (green) and inhibitory (red) balance (**D**). After decades and with aging, the motor system fails as further risk factors take effect in the multistep process hypothesized for ALS to become symptomatic and relentlessly progressive (**E**).

**Table 1 brainsci-15-00195-t001:** Mechanisms by which neonatal dysbiosis could impact neurodegeneration.

1	Impaired immune system programming
2	Miscommunication through the gut–brain axis
3	Metabolite toxicity
4	Blood–brain barrier breakdown
5	Epigenetic modulation
6	Mitochondrial dysfunction
7	Misfolded protein aggregation
8	Dysregulation of the hypothalamic–pituitary–adrenal axis
9	Altered neurotransmitter production

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
