# Peer review of "The Neonatal Microbiome: Implications for Amyotrophic Lateral Sclerosis and Other Neurodegenerations"

_brainsci, 2025, doi:10.3390/brainsci15020195_

Round 1
Reviewer 1 Report
Comments and Suggestions for Authors
This is a useful introduction to early micrbiome changes and the risk of ALS, which one would consider a viable avenue for further research.
Drawing attention to the vulnerability of the infant gut micrbiome is very helpful more generally.
I would suggest a brief account of ALS pathology in the introduction. Otherwise the paper is a useful and interesting read.
A few grammatical issues:
Line 148 "Inflammatory control is mediated through microglia the immune cells of the brain parenchyma". ?hyphen between microglia and the.
Line 156 - 'wich' should read 'which'
Line 213/214 needs revision (perhaps lose the "As the")
Author Response
Reviewer 1
This is a useful introduction to early micrbiome changes and the risk of ALS, which one would consider a viable avenue for further research.
Drawing attention to the vulnerability of the infant gut micrbiome is very helpful more generally.
I would suggest a brief account of ALS pathology in the introduction. Otherwise the paper is a useful and interesting read.
A few grammatical issues:
Line 148 "Inflammatory control is mediated through microglia the immune cells of the brain parenchyma". ?hyphen between microglia and the.
Line 156 - 'wich' should read 'which'
Line 213/214 needs revision (perhaps lose the "As the")
RESPONSE
We thank the reviewer for their comments and suggestions. As suggested a paragraph has been added to the introduction regarding the pathophysiology of ALS.
The grammatical errors have been corrected.
Reviewer 2 Report
Comments and Suggestions for Authors
The review by Eisen and Kiernan is well written but the driving hypothesis seems far fetched. The authors, both experts in the field, do not refer to their research rather they focus on something far more elusive: neonatal microbiome. I do not quite grasp why changes in neonatal microbiome would be implicated in the pathogenesis of ALS, and not other neurodegenerative diseases? Isn't it too narrow? I am aware that diet plays a role in ALS, but is it possible that neonatal microbiome single handedly could have such a dramatic role in pathogenesis of ALS? And if so, what could we do to make the situation better?
I think that adding more figures with clear explanation of signaling pathways would definitely benefit the paper. Authors might also consider expanding the title by adding: and other neurodegenerative diseases.
There are a couple of minor language issues: wich instead of which; mental shortcuts; "left-over" sentences under the main text i.e. Funding: Please add: “This research received no external funding”. 324
Institutional Review Board Statement: “Not applicable” for studies not involving humans or animals. Informed Consent Statement: “Not applicable.” for studies not involving humans. You might also choose to exclude this statement if the study did not involve humans.
Author Response
Reviewer 2
The review by Eisen and Kiernan is well written but the driving hypothesis seems far fetched. The authors, both experts in the field, do not refer to their research rather they focus on something far more elusive: neonatal microbiome. I do not quite grasp why changes in neonatal microbiome would be implicated in the pathogenesis of ALS, and not other neurodegenerative diseases? Isn't it too narrow? I am aware that diet plays a role in ALS, but is it possible that neonatal microbiome single handedly could have such a dramatic role in pathogenesis of ALS? And if so, what could we do to make the situation better?
I think that adding more figures with clear explanation of signaling pathways would definitely benefit the paper. Authors might also consider expanding the title by adding: and other neurodegenerative diseases.
There are a couple of minor language issues: wich instead of which; mental shortcuts; "left-over" sentences under the main text i.e. Funding: Please add: “This research received no external funding”. 324
Institutional Review Board Statement: “Not applicable” for studies not involving humans or animals. Informed Consent Statement: “Not applicable.” for studies not involving humans. You might also choose to exclude this statement if the study did not involve humans.
RESPONSE
We thank the reviewer for the comments and suggestions. In response we have tried where applicable to emphasize that the concept we are proposing is one in the multistep process of ALS, in particular as relating to step one early life. The figure has been modified to incorporate this. We do not feel that additional figures will be helpful. We have added other neurodegenerations to the title. We have addressed the grammatical errors.
Round 2
Reviewer 2 Report
Comments and Suggestions for Authors
The responses are satisfactory, I recommend publishing the manuscript